# Optical-Fiber Microsphere-Based Temperature Sensors with ZnO ALD Coating—Comparative Study

**DOI:** 10.3390/s21154982

**Published:** 2021-07-22

**Authors:** Paulina Listewnik, Mikhael Bechelany, Paweł Wierzba, Małgorzata Szczerska

**Affiliations:** 1Department of Metrology and Optoelectronics, Faculty of Electronics, Telecommunications and Informatics, Gdańsk University of Technology, 11/12 Narutowicza Street, 80-233 Gdańsk, Poland; pwierzba@eti.pg.edu.pl; 2Institut Europeén Des Membranes, IEM, UMR 5635, Univ of Montpellier, CNRS, ENSCM, CEDEX 5, 34095 Montpellier, France; mikhael.bechelany@umontpellier.fr

**Keywords:** temperature sensor, fiber-optic sensor, photonic sensor, atomic layer deposition, microsphere, temperature, ZnO

## Abstract

This study presents the microsphere-based fiber-optic sensor with the ZnO Atomic Layer Deposition coating thickness of 100 nm and 200 nm for temperature measurements. Metrological properties of the sensor were investigated over the temperature range from 100 °C to 300 °C, with a 10 °C step. The interferometric signal was used to monitor the integrity of the microsphere and its attachment to the connecting fiber. For the sensor with a 100 nm coating, a spectrum shift of the reflected signal and the optical power of the reflected signal were used to measure temperature, while only the optical power of the reflected signal was used in the sensor with a 200 nm coating. The R^2^ coefficient of the discussed sensors indicates a linear fit of over 0.99 to the obtained data. The sensitivity of the sensors, investigated in this study, equals 103.5 nW/°C and 19 pm/°C or 11.4 nW/°C for ZnO thickness of 200 nm and 100 nm, respectively.

## 1. Introduction

Fiber-optic sensors have been developed and improved upon for a few decades. Due to their versatility, they are used in numerous fields, in industry, science and medicine [1,2,3,4]. Because of their many advantages, such as chemical inertness or resistance to electromagnetic interference, they can often be utilized in places where electric sensors cannot be applied, e.g., in an explosive or combustible atmosphere, severe climatic conditions and hard to access places [5,6,7,8]. Optimization of measurement parameters plays a significant role in the development of the fiber-optic sensors. While planning measurements, the selection of the sensor is a crucial element, depending on the application and conditions in which they will be performed. There are many ways to tune metrological properties of fiber-optic sensors.

Probably the most common method is modification of the sensor’s structure. A broad variety of sensing heads can be created thanks to advancements in technology and fabrication techniques—especially in fiber-optic fusion splicers and laser-splicing systems, femtosecond lasers and chemical etching [9,10,11,12,13]—that allow a multitude of fiber-optic structures. Many of them, including microdiscs, microrings, tapers, microresonators and microspheres, are coupled with a high coherent light source to generate resonance within the structure, on the basis of the Whispering Mode Gallery (WMG) phenomenon [14,15]. 

Another way to modify fiber-optic sensors is by addition of a coating on either the flat end-face of the fiber or on the surface of the structure. The coatings consist of a variety of materials, mostly 2D, and they can be deposited in different forms. Based on their properties they are used for specific applications. For example, due to its structure, graphene oxide coating is ideal for humidity and refractive index measurements [16]. Single-walled carbon nanotubes (SWCNT) [17] are used for the detection of ammonia, ethanol and methanol vapor because of their interaction with the coating. The Pd-Au layer is used for hydrogen-concentration sensing [18].

The deposition method of the coating is an important issue while modifying parameters of fiber-optic sensors. Depending on the deposition technique, e.g., dip-coating method, Atomic Layer Deposition or magnetron sputtering, geometry, uniformity and conformity vary widely [19,20,21,22,23,24].

Based on the type of sensor, several parameters can be optimized, e.g., adjustable cavity length, structure modification [25,26,27,28] and modifying metrological properties such as resolution, precision, sensitivity and accuracy [29,30]. Many researchers contribute to determining the properties and parameters of various materials and structures [31,32].

This study investigates the performance of the microsphere-based fiber-optic temperature sensors with a 100 nm and 200 nm ZnO ALD coating. By combining several methods of modification of the fiber-optic sensors, the complexity of the measurement system can be reduced and the number of adjustable parameters is increased. The ZnO sensor properties were investigated in the range of 100 °C to 300 °C. In this temperature range, ZnO is well known for its chemical stability in a broad variety of gaseous and liquid environments. For this reason, ZnO is widely used for gas-sensor applications, for instance [33].

## 2. Materials and Methods

Measurements were performed using a sensor made of a standard single-mode telecommunication optical fiber (SMF-28, Thorlabs Inc., Newton, NJ, USA) with a microsphere structure produced at the end of the fiber, using a fiber-optic splicer (FSU975, Ericsson, Stockholm, Sweden). To obtain a microsphere with a diameter of 245 µm, splicing proceeded in a three-step pull. The fiber-optic splicer uses an electric arc to melt the fiber, while the pulling causes the formation of a microsphere. By varying splicing parameters, such as time of pull, electric arc current and fiber distance, the process can be controlled to achieve a highly repeatable structure. After the manufacturing of the microsphere, the ZnO coating was deposited on its surface by the Atomic Layer Deposition (ALD) method. ALD is a vapor phase deposition technique enabling the synthesis of ultrathin films with a sub-nanometer thickness control. A key benefit of ALD is high conformality of produced layers. In fact, ALD can be used to coat complex 3D substrates with a conformal and uniform layer of high-quality materials, a capability unique amongst thin film deposition techniques. Therefore, by using ALD, we were certain that ZnO would be uniformly and accurately grown on the microspheres inside the deposition chamber [34]. A custom-made ALD reactor was used for deposition of ZnO layers. ALD was performed using sequential exposures of Diethyl Zinc and H_2_O, separated by a purge of nitrogen with a flow rate of 100 sccm. The deposition regime for ZnO consisted of a 0.1 s pulse of DEZ, 20 s of exposure to DEZ, 40 s of purge with argon followed by 2 s pulse of H_2_O, 30 s of exposure to H_2_O and finally 60 s purge with argon. ZnO thin films with a different number of cycles were deposited on the microsphere and on Si substrates, for reference purposes. The temperature during the process was fixed at 100 °C [35,36]. After the deposition process, the sensor was connected to an optical coupler by fusion splicing.

To assess the quality of the structure and the deposited ZnO ALD coating of 100 nm thickness, the microsphere was investigated under Scanning Electron Microscope (SEM, Phenom XL G2, Thermo Fisher Scientific, Waltham, MA, USA). An example image of the microsphere, recorded at 1000x magnification, is shown in Figure 1.

The structure presented in Figure 1 exhibits excellent roundness, and the presence of the ZnO coating is confirmed.

The metrological properties of the sensor were assessed by performing measurements. The utilized setup, which consists of a light source, an optical spectrum analyzer, the developed sensor and a temperature calibrator, is presented in Figure 2.

During the investigation, the sensor was placed in the temperature calibrator (ETC-400A, Ametek, Berwyn, PA, USA). Temperature was increased from 100 °C to 300 °C, with a 10 °C step. At each step the measurement was performed three minutes after the temperature had stabilized, allowing the sensor to adjust to altered conditions. The measurements were executed using a light source with a center wavelength of 1310 nm ± 20 nm (SLD-1310-18-W, FiberLabs Inc., Fujimino, Japan). The signal was propagated through a 2:1 50/50% optical coupler (G657A, CELLCO, Kobylanka, Poland) to the sensor head coated with a 100 nm ZnO ALD coating, where it was reflected, as shown in Figure 3. Due to differences in the refractive indices, the incident signal was reflected principally from the boundary between the core and the cladding glass, as well as from the boundary between the cladding glass and the ZnO layer. Both reflected waves were superposed and detected by the Optical Spectrum Analyzer (Ando AQ6319, Yokohama, Japan), in which they interfered, giving rise to a spectral interference pattern

The presence of the interference pattern confirms the integrity of the structure, ensuring that the sensor was not damaged. Depending on the position of the spectral peak of the signal, the temperature can be determined.

## 3. Results and Discussion

This section presents results that were acquired from the measurements performed with the setup shown above for the sensor with a 100 nm ZnO ALD coating. The detailed results of an investigation of the sensor with 200 nm coating are presented elsewhere [37]. This section also describes the comparison of the data sets obtained for both sensors. The following comparison does not intend to determine which sensor or analysis type is better, but to present their metrological parameters.

Figure 4 shows normalized values of the measured signal response for the microsphere-based sensor with a 100 nm ZnO ALD coating at 100 °C and 300 °C to preserve the readability of the plot. By increasing the temperature, the spectral peak of the reflected signal shifts toward lower values of the wavelength. In addition, interference modulation of the spectrum visible in Figure 4 confirms the integrity of the sensor head structure, allowing its condition to be monitored in real time.

The dependence of the peak wavelength position on the temperature is shown in Figure 5, along with the linear fit to the data. Coefficient R^2^ equals 0.991, confirming good fit of the obtained data to the theoretical model. Furthermore, sensitivity of the microsphere-based sensor with a 100 nm ZnO ALD coating was calculated to be 19 pm/°C.

The spectrum changes its peak wavelength position when the temperature is altered. As the temperature rises, the spectrum shifts by a constant value throughout the whole range of roughly 2 nm per 100 °C. By following linear regression, it is possible to determine the measured temperature using the position of the peak of the reflected signal.

In addition to the peak wavelength shift, the sensor with a 100 nm ZnO ALD coating also retains the same property as the sensor with a 200 nm coating. The measured response of the reflected signal optical power for the sensor with a 100 nm coating is presented in Figure 6.

The peak optical power of the reflected signal increases when the temperature rises. The dependence of the reflected signal’s optical power on the temperature is presented in Figure 7.

Based on the results presented in Figure 7, it can be concluded that the increase of the signal throughout the examined range is 16%. The sensitivity of the microsphere-based sensor with 100 nm ZnO ALD coating was also calculated, and it equals 11.35 nW/°C. A linear fit to the experimental data was included in Figure 7 and the coefficient R^2^ = 0.999 was calculated, confirming the good fit of obtained data to the theoretical model.

In addition, the values of linearity error, sensitivity error and approximation error were calculated. The data for the sensor with a 100 nm coating include both spectral and optical power analysis, because the device provides two ways of measurement.

Based on acquired data, the sensitivity of the sensors and sensitivity error, a deviation of this parameter from the one fitted from the theoretical zero-deviation slope, was calculated respectively from the Formulas (1) and (2):(1)$$S=\frac{\Delta P}{\Delta T},\;$$
where: *S*—sensitivity of the sensor, Δ*P*—optical power of the reflected signal, Δ*T*—temperature range.

The sensitivity of the sensor with optical power analysis cannot be compared to the one with spectral shift. However, the parameters of both analysis methods are shown to provide metrological capabilities of the sensor. Depending on the nature of designed measurements, the required sensitivity can differ.
(2)$$u_{sensitivity}=\left(1-\frac{S_{theor}}{S}\right)*100\%$$
where: *u_sensitivity_*—sensitivity error, *S_theor_*—theoretical fitting sensitivity.

As observed, in the spectral shift analysis the sensitivity error equals 0%. The offset of the values is present, but the relative deviation takes the same values on both sides of the range. The sensitivity error in optical power analysis is close to 1%.

The dependence of the changes occurring in the spectra on increasing temperature during measurements with each sensor is consistent with the linear characteristic, and the linear fit was included in the graphs. Therefore, the uncertainty of the obtained data to the theoretical model was calculated (3):(3)$$u_{linearity}=\frac{\max\left|P-P_{theor}\right|}{\Delta I},$$
where: *u_linearity_*—linearity error, *P*—optical power of the reflected signal, *P_theor_*—theoretical fitting optical power.

The linearity error is almost 5 times lower while analyzing the data of optical power, in comparison to those analyzed in the spectral shift analysis.

The next calculated parameter was approximation error, which is the highest normalized difference between obtained data and its linear fit. The approximation error was calculated from the following Formula (4):(4)$$\delta =\left|\frac{\upsilon_{M}-\upsilon_{R}}{\upsilon_{R}}\right|*100\%,$$
where: *δ*—approximation error, *υ_M_*—obtained data, *υ_R_*—linear fit.

For the approximation error, while in both instances it remains below 0.2%, its value is 10 times lower (0.02%) with spectral shift analysis. The parameters are presented in Table 1.

Based on the R^2^ coefficient, it can be determined that the microsphere-based sensor with optical power analysis exhibits higher linearity.

Comparison of the normalized spectra of reflected signals from sensors with 200 nm and 100 nm ZnO coating at 100 °C is presented in Figure 8. Normalization was performed to the highest value of optical power. It can be observed that the optical power of a sensor with a 200 nm coating is over 10 times higher than the optical power of a sensor with a 100 nm coating

Figure 9a shows the dependence of the peak optical power on temperature for micro-sphere fiber-optic sensors and their difference while using 200 nm and 100 nm ZnO ALD coating. Figure 9b also shows their theoretical linear fit described by a linear function. Because the linear fit is a close match to the obtained data, by following linear regression it is possible to accurately predict metrological properties of the sensor at a temperature beyond the investigated range and to forecast the behavior of the sensor with any other thickness of a ZnO ALD coating. By considering the slope of the function, the sensitivity of the sensor can be further estimated, whereas by analyzing the y-intercept, the optical power level can be estimated.

In Table 2, a comparison of the metrological parameters for both sensors is presented.

As mentioned before, the measurement properties of the microsphere-based sensors with ZnO ALD coatings of 100 nm and 200 nm were investigated in the temperature range of 100 °C–300 °C.

While the sensitivity error of the sensor with a 100 nm coating is, at most, 1%, the sensor with a 200 nm coating exhibits a 3.97% error. Throughout measurement planning, the sensitivity needed to accomplish the task must be considered. For investigations requiring lower sensitivity, the sensor with thinner ZnO ALD coating may still provide sufficient metrological parameters, while having a clear advantage in terms of time and cost of its production. As can be seen, the linearity error in terms of spectral analysis is almost 5 times lower for the sensor with a 100 nm coating. However, considering its linearity error by spectral analysis, it also deviates by 5%. For both presented sensors, the approximation error was less than 0.5%. By comparing R^2^ coefficients of both sensors, it is shown that the microsphere-based fiber-optic sensor with a 100 nm ZnO ALD coating indicates a closer match of the obtained data to the theoretical fit, and therefore its measured characteristic exhibits better linearity than that of the sensor with a 200 nm coating.

## 4. Conclusions

Microsphere-based sensors are good for long-term and remote measurement of temperature as integrity of the sensor head can be monitored, allowing any failure of the sensing head to be reliably detected. In this study, performance of a microsphere-based fiber-optic sensor for temperature measurements with a 100 nm ZnO layer was investigated and compared with that of a sensor with a 200 nm ZnO layer. The sensors exhibited a close match between measurement data and theoretical linear fit, which is confirmed by an R^2^ coefficient exceeding 0.99. The sensitivity of the sensor with a 100 nm coating equaled 0.019 nm/°C, while the sensitivity of the sensor with a 200 nm coating equaled 103.5 nW/°C. Additionally, for the microsphere-based sensor with a 100 nm ZnO ALD coating, changes of temperature were observed based on the optical power increase, which coincided with the rise of the temperature.

## Figures and Tables

**Figure 1 sensors-21-04982-f001:**
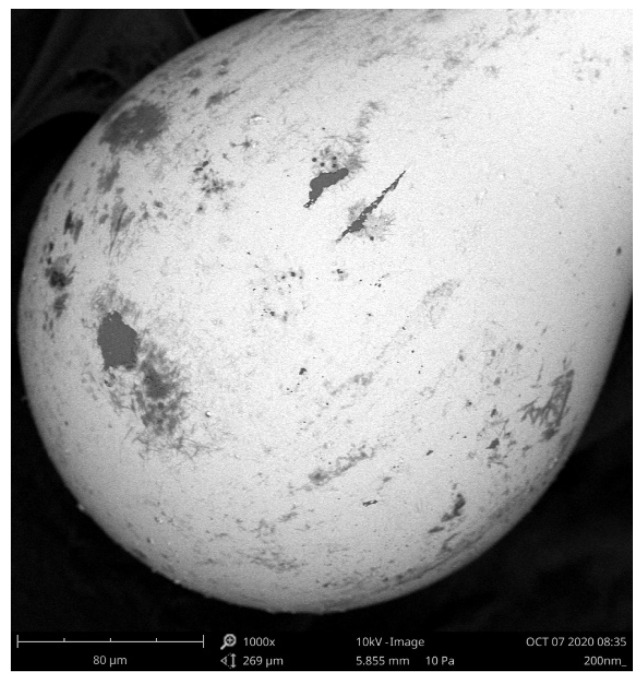
SEM image of the microsphere sensor with a 100 nm ZnO ALD coating. Magnification of 1000×.

**Figure 2 sensors-21-04982-f002:**
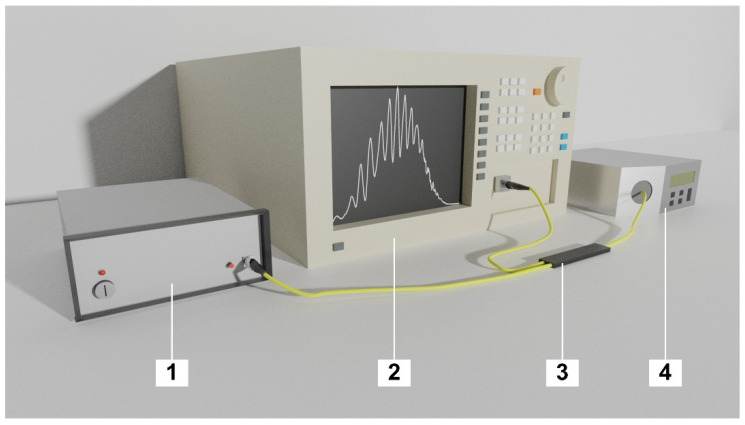
Experimental setup, where: 1—light source, 2—Optical Spectrum Analyzer, 3—Optical fiber coupler, 4—Temperature calibrator.

**Figure 3 sensors-21-04982-f003:**
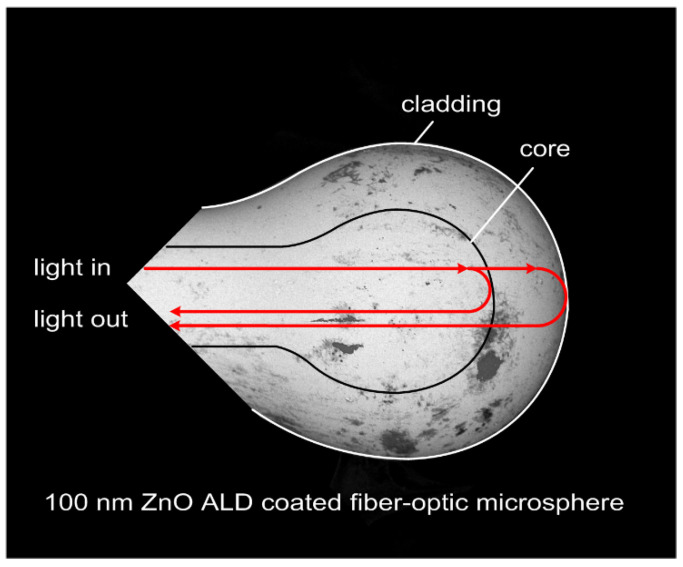
Principle of operation of a microsphere-based fiber-optic sensor.

**Figure 4 sensors-21-04982-f004:**
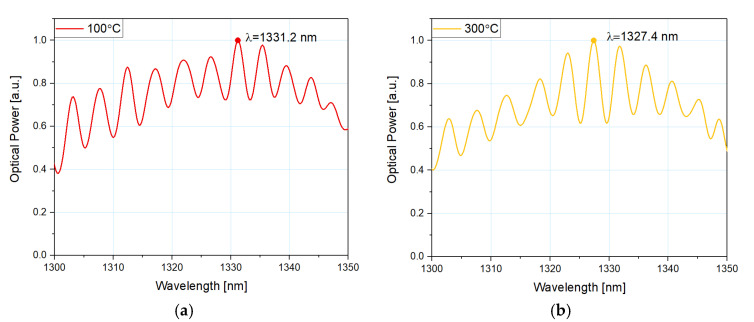
Normalized measured response of the reflected signal for the microsphere-based sensor with 100 nm ZnO ALD coating at (**a**) 100 °C and (**b**) 300 °C.

**Figure 5 sensors-21-04982-f005:**
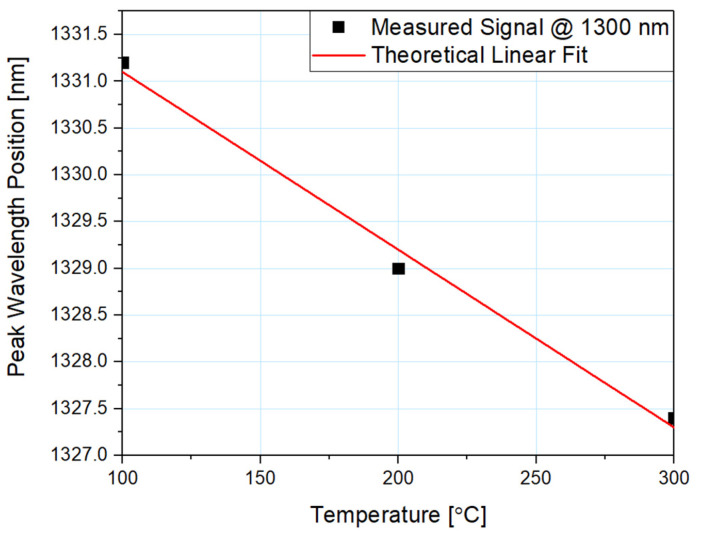
Dependence of the spectral shift of a reflected signal on the temperature.

**Figure 6 sensors-21-04982-f006:**
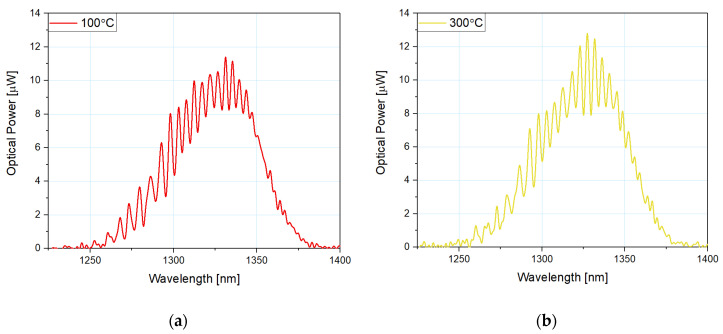
Measured response of the reflected signal optical power for the microsphere-based fiber-optic temperature sensor at: (**a**) 100 °C, (**b**) 300 °C.

**Figure 7 sensors-21-04982-f007:**
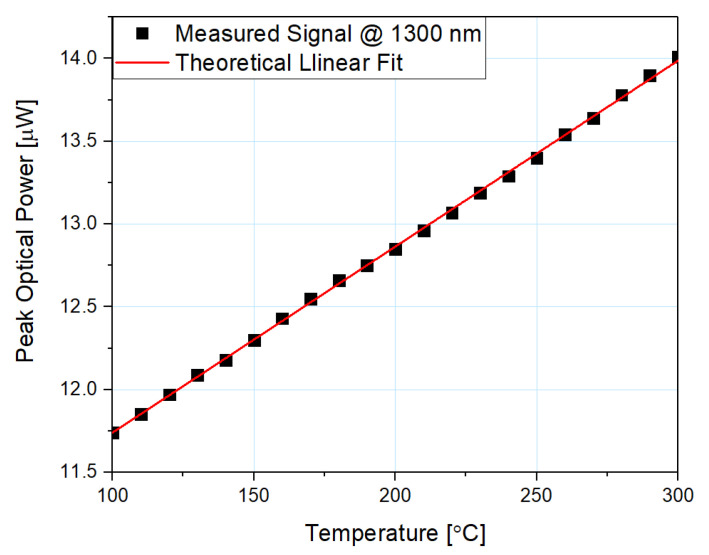
Reflected signal peak optical power dependence on the changing temperature and its theoretical linear fit, measured at a wavelength of 1300 nm, using the microsphere-based sensor with 100 nm ZnO ALD coating.

**Figure 8 sensors-21-04982-f008:**
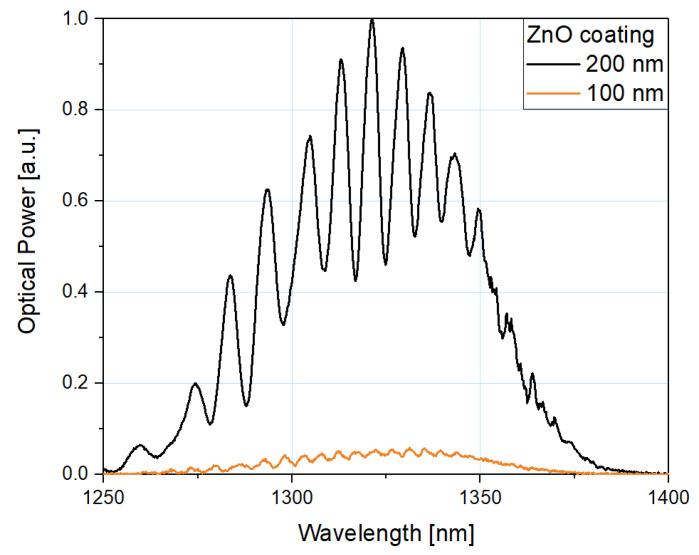
Normalized reflected spectra measured at 100 °C with microsphere-based sensors with a 200 nm (black line) and a 100 nm (orange line) ZnO ALD coating.

**Figure 9 sensors-21-04982-f009:**
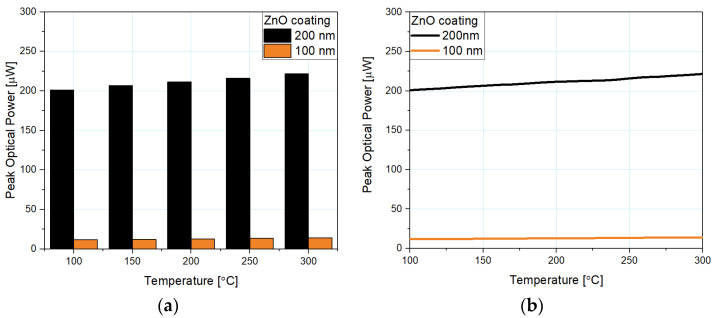
The dependency of the peak optical power on the temperature: (**a**) comparison of the obtained peak optical power of the reflected signal for each sensor, (**b**) obtained data and linear approximation.

**Table 1 sensors-21-04982-t001:** Comparison between optical power and spectral shift analysis of parameters of the microsphere-based sensor with a 100 nm coating.

Parameter	100 nm Coating	
Investigated range [°C]	100–300	
Characteristics	linear	
Analysis type	spectral shift	optical power
Sensitivity	19 pm/°C	11.35 nW/°C
Sensitivity error [%]	0	0.99
Theoretical sensitivity [%]	19 pm/°C	11.24 nW/°C
Linearity error [%]	5.2	1.15
Approximation error [%]	0.02	0.2
R^2^	0.992	0.999

**Table 2 sensors-21-04982-t002:** List of metrological parameters of the microsphere-based fiber-optic temperature sensors—optical power analysis.

Parameter	100 nm Coating	200 nm Coating
Investigated range [°C]	100–300	
Characteristics	linear	
Analysis type	optical power	
Sensitivity [nW/°C]	11.35	103.5
Sensitivity error [%]	0.99	3.97
Theoretical sensitivity [nW/°C]	11.24	99.39
Linearity error [%]	1.15	5
Approximation error [%]	0.2	0.49
R^2^	0.999	0.995

## Data Availability

The data presented in this study are openly available in MOST Wiedzy repository at https://doi.org/10.34808/3g4h-7p44 (accessed on 21 July 2021).

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
