# Peer review of "Optical-Fiber Microsphere-Based Temperature Sensors with ZnO ALD Coating—Comparative Study"

_sensors, 2021, doi:10.3390/s21154982_

Round 1
Reviewer 1 Report
You have improved the manuscript, but it is not yet satisfactory for me. For this, my recommendation is major review.
The title and focus of the manuscript were changed to “comparative study”, with that You moved away from the problem of good power measurement faced in intensity sensors, but You keep saying on the manuscript that spectral measurements are only to check for the integrity of the sensor. On the other hand, for a complete comparison between the two different thickness, I would expect that a spectral analysis with temperature would be performed also for the 200 micron thickness, and a comparison made (as has made for the 100 micron).
Usually the great advantage of intensity sensors is the simplicity of the detection setup (only one, or two detectors). For the measurement of peak power, as done, You need to include a spectral analysis of the output. Another problem that I envisage is the fringe modulation that can be seen in figure 4 (due to more than one FP cavity), this will influence surely the linearity of the peak power measurements.
A few more comments:
Fig 5 – experimental points should be visible. The line presented (two straight segments) is confusing. The same should apply for figure 7, were we can see more straight lines.
Lines 140-141 – “By following linear regression, it is possible to determine the position of the reflected signal peak for each measured temperature.” The objective should be the opposite (measuring temperature for each peak position)
The English usage should be checked. There are a lot of improvements in the phrase construction that could improve the reading of the manuscript.
Author Response
The title and focus of the manuscript were changed to “comparative study”, with that You moved away from the problem of good power measurement faced in intensity sensors, but You keep saying on the manuscript that spectral measurements are only to check for the integrity of the sensor. On the other hand, for a complete comparison between the two different thickness, I would expect that a spectral analysis with temperature would be performed also for the 200 micron thickness, and a comparison made (as has made for the 100 micron).
Thank you for this comment. The sensor with a 200 nm ZnO ALD coating exhibits a spectral shift that makes spectral analysis difficult.
Usually the great advantage of intensity sensors is the simplicity of the detection setup (only one, or two detectors). For the measurement of peak power, as done, You need to include a spectral analysis of the output. Another problem that I envisage is the fringe modulation that can be seen in figure 4 (due to more than one FP cavity), this will influence surely the linearity of the peak power measurements.
Thank you for this comment. The fringe modulation, visible in Figure 4, confirms the integrity of the sensor structure. It can be monitored either in real-time or periodically, at intervals ranging from seconds to days. The monitoring does not require full spectral analysis, as it potentially can be performed using 2 – 3 laser diodes operating at suitably chosen wavelengths, and a photodiode acting as the detector. Fringe modulation introduced by the Fabry-Perot cavities present in the structure of the sensor is going to be the subject of further research.
Fig 5 – experimental points should be visible. The line presented (two straight segments) is confusing. The same should apply for figure 7, were we can see more straight lines.
Requested changes were introduced.
Lines 140-141 – “By following linear regression, it is possible to determine the position of the reflected signal peak for each measured temperature.” The objective should be the opposite (measuring temperature for each peak position)
Thank you for noticing this, rather unfortunate, statement. It was rectified to read “By following linear regression, it is possible to determine the measured temperature using the position of the peak of the reflected signal” .
The English usage should be checked. There are a lot of improvements in the phrase construction that could improve the reading of the manuscript.
The language of the manuscript was checked and corrections were introduced wherever needed.
Reviewer 2 Report
1. Title does not describe properly the content of this manuscript. Especially, "photonic" might mislead to other contents. Authors should re-consider the title, itself. 2. The scientific investigation is not clear in this manuscript. 3. The usefulness of these temperature sensors is not clear. What kind of application can you expect? 4. They published already a similar result in the same journal. The new findings in this manuscript after that is not clear.Author Response
- Title does not describe properly the content of this manuscript. Especially, "photonic" might mislead to other contents. Authors should re-consider the title, itself.
While there was no objection raised in the whole editorial process to date regarding the word “photonic”, the title was modified to “Optical-fibre microsphere-based temperature sensors with ZnO ALD coating – comparative study”
- The scientific investigation is not clear in this manuscript.
The corrections were introduced in the manuscript that should make it more clear.
- The usefulness of these temperature sensors is not clear. What kind of application can you expect?
Target applications are mostly in industrial monitoring, in applications where electrical sensors are to be avoided and where moderate temperature resolution and accuracy are needed, and the ability to verify the integrity of the sensor is required, or at least beneficial.
- They published already a similar result in the same journal. The new findings in this manuscript after that is not clear.
Indeed, the results of the previous investigation were published in Sensors (Sensors 2020, 20, 4689; doi:10.3390/s20174689). In the current manuscript, the investigation is extended to a sensor with a 100-nm layer of ZnO and covers a comparison of sensors with a 100-nm layer of ZnO and a 200-nm layer of ZnO.
Reviewer 3 Report
The author used the atomic layer depositon method to coat the ZnO fiber sensor with the microsphere structure for temperature sensing, and comparedt the sesing effect when the thickness of the coated ZnO was 100nm and 200nm, by analaysing the changes of peak wavelength positon and optical power. In addition, the author analyzed several errors and performed a linear fit. Finally, it is concluded that the ALD method for coating ZnO for temperature measurement has a good result. However, the paper can be considered for publication provided the following concerns are addressed:
- Regarding material selection, why choose ZnO? please explain in detail waht are the reasons for choosing ZnO.
- For the choice of ZnO thickness, only 100nm and 200nm were selected. What is the basis for selecting these two paraments?
- What are the advantages of the device proposed by the author, especially in terms of sensitivity.
- It only explores the structure's sensing of temperature when the temperature rises. Is it enough to show the reusability of the sensing structure? Why not consider the structure's sensing of temperature when the temperature drops?
- What is the relationship between Figure 4 and Figure 6? Is Fig.4 a localized normalization of Fig.6? If so, what is the significant of doing this. If not, please explain in detail the meaning of the two pictures.
- What is the relationship between graph a) and graph b) in Figure 9? Is it just the difference between a histogram and a linear graph? Should their ordinates "Peak Intensity[μW]" and "Peak Optical Power[μW]" be consistent?
7. I suggest a check on the English language.

Author Response
- Regarding material selection, why choose ZnO? please explain in detail waht are the reasons for choosing ZnO.
ZnO coating was chosen because it offered a couple of advantages in our case. First, zinc oxide nanomaterials have a thermal expansion coefficient greater by two orders of magnitude over pure silica – about [1]. Second, the refractive index of ZnO changes substantially with temperature [2]. Third, ZnO is relatively resistant to chemical aggression and temperature. Fourth, using Atomic Layer Deposition to deposit ZnO results in conformal layers of this material with homogenous structure, dimensions, and properties. Finally, it should be noted that the properties of ZnO layers can be tuned in a certain range, as described in [3]. All these characteristics make ZnO a good candidate for a sensing layer in optical fiber sensors.
References:
- Singh, M.; Singh, M. Thermal Expansion in Zinc Oxide Nanomaterials. Nanoscience and Nanotechnology Research 2013, 1, 27–29, doi:10.12691/nnr-1-2-4.
- Qiu, K.; Zhao, Y.; Gao, Y.; Liu, X.; Ji, X.; Cao, S.; Tang, J.; Sun, Y.; Zhang, D.; Feng, B.; et al. Refractive Index of a Single ZnO Microwire at High Temperatures. Appl. Phys. Lett. 2014, 104, 081109, doi:10.1063/1.4866668.
- Pilz, J.; Perrotta, A.; Christian, P.; Tazreiter, M.; Resel, R.; Leising, G.; Griesser, T.; Coclite, A.M. Tuning of Material Properties of ZnO Thin Films Grown by Plasma-Enhanced Atomic Layer Deposition at Room Temperature. Journal of Vacuum Science & Technology A 2017, 36, 01A109, doi:10.1116/1.5003334.
- For the choice of ZnO thickness, only 100nm and 200nm were selected. What is the basis for selecting these two paraments?
Atomic Layer Deposition, in spite of its advantages, is a time-consuming technique, with deposition cost increasing rapidly with increasing thickness of the layer. Therefore, the lowest possible thickness of the layer yielding adequate performance is usually seen as the optimal choice. During preliminary research on the sensors, a range of structures with different thickness values of the ZnO coating were tested. Structures with the coating thickness up to 70 nm did not produce an acceptable response, while structures with coating thickness over 300 nm did not offer any substantial performance benefits, requiring prolonged deposition times.
- What are the advantages of the device proposed by the author, especially in terms of sensitivity?
The devices discussed in the manuscript were not developed with a view to attaining the highest possible sensitivity. If we were to develop such a sensor, a structure with a high-quality factor resonator, such as a microsphere [4], would be our first choice. The advantages of presented sensors are the ability to verify their integrity during operation, simplicity, low cost, and ability to work where electrical sensors are not allowed or their performance is affected by Electromagnetic Interference (EMC). The sensitivity demonstrated in this paper is 13.37 pm/°C, and is of the same order as that of Fibre Bragg Grating sensors (c.a. 10 pm/°C).
References:
- Soria, S.; Berneschi, S.; Brenci, M.; Cosi, F.; Nunzi Conti, G.; Pelli, S.; Righini, G.C. Optical Microspherical Resonators for Biomedical Sensing. Sensors 2011, 11, 785–805, doi:10.3390/s110100785.
- It only explores the structure's sensing of temperature when the temperature rises. Is it enough to show the reusability of the sensing structure? Why not consider the structure's sensing of temperature when the temperature drops?
Hysteresis of any sensing device is indeed an important problem. The sensors were tested to return to the starting point upon cooling and did not show any deviation from its original recorded characteristics.
- What is the relationship between Figure 4 and Figure 6? Is Fig.4 a localized normalization of Fig.6? If so, what is the significant of doing this. If not, please explain in detail the meaning of the two pictures.
Figure 4 was normalized and magnified at the values of a central wavelength to accentuate and focus on the shift of the spectrum, which is not clearly visible in Figure 6.
- What is the relationship between graph a) and graph b) in Figure 9? Is it just the difference between a histogram and a linear graph? Should their ordinates "Peak Intensity[μW]" and "Peak Optical Power[μW]" be consistent?
Yes, in both cases the axes should be titled as “Peak Optical Power [µm]”, it has been corrected.
- I suggest a check on the English language.
The language of the manuscript was checked and corrections were introduced wherever needed.
Round 2
Reviewer 1 Report
I am satisfied with the revision made.
Author Response
Dear Reviewer,
we are happy that you are satisfied with our answers.
Best regards
Authors
Reviewer 3 Report
The authors' have satisfactorily addressed my comments and made the changes to the revised manuscript. This manuscript can be considered for publication.
Author Response

(The authors gave the same response as above.)
